# The Combined Effect of Polygenic Risk Score and Prostate Health Index in Chinese Men Undergoing Prostate Biopsy

**DOI:** 10.3390/jcm12041343

**Published:** 2023-02-08

**Authors:** Xiaohao Ruan, Da Huang, Jingyi Huang, Jinlun Huang, Yongle Zhan, Yishuo Wu, Qiang Ding, Danfeng Xu, Haowen Jiang, Wei Xue, Rong Na

**Affiliations:** 1Department of Urology, Ruijin Hospital, Shanghai Jiao Tong University School of Medicine, Shanghai 200025, China; 2Division of Urology, Department of Surgery, The University of Hong Kong, Hong Kong, China; 3Department of Urology, Huashan Hospital, Fudan University, Shanghai 200040, China; 4State Key Laboratory of Oncogenes and Related Genes, Department of Urology, Renji Hospital, Shanghai Jiao Tong University School of Medicine, Shanghai 200127, China

**Keywords:** polygenic risk score, prostate biopsy, prostate cancer, prostate health index, prostate-specific antigen

## Abstract

To date, the combined effect of polygenic risk score (PRS) and prostate health index (*phi*) on PCa diagnosis in men undergoing prostate biopsy has never been investigated. A total of 3166 patients who underwent initial prostate biopsy in three tertiary medical centers from August 2013 to March 2019 were included. PRS was calculated on the basis of the genotype of 102 reported East-Asian-specific risk variants. It was then evaluated in the univariable or multivariable logistic regression models that were internally validated using repeated 10-fold cross-validation. Discriminative performance was assessed by area under the receiver operating curve (AUC) and net reclassification improvement (NRI) index. Compared with men in the first quintile of age and family history adjusted PRS, those in the second, third, fourth, and fifth quintiles were 1.86 (odds ratio, 95% confidence interval (CI): 1.34–2.56), 2.07 (95%CI: 1.50–2.84), 3.26 (95%CI: 2.36–4.48), and 5.06 (95%CI: 3.68–6.97) times as likely to develop PCa (all *p* < 0.001). Adjustment for other clinical parameters yielded similar results. Among patients with prostate-specific antigen (PSA) at 2–10 ng/mL or 2–20 ng/mL, PRS still had an observable ability to differentiate PCa in the group of prostate health index (*phi*) at 27–36 (*P*_trend_ < 0.05) or >36 (*P*_trend_ ≤ 0.001). Notably, men with moderate *phi* (27–36) but highest PRS (top 20% percentile) would have a comparable risk of PCa (positive rate: 26.7% or 31.3%) than men with high *phi* (>36) but lowest PRS (bottom 20% percentile positive rate: 27.4% or 34.2%). The combined model of PRS, *phi*, and other clinical risk factors provided significantly better performance (AUC: 0.904, 95%CI: 0.887–0.921) than models without PRS. Adding PRS to clinical risk models could provide significant net benefit (NRI, from 8.6% to 27.6%), especially in those early onset patients (NRI, from 29.2% to 44.9%). PRS may provide additional predictive value over *phi* for PCa. The combination of PRS and *phi* that effectively captured both clinical and genetic PCa risk is clinically practical, even in patients with gray-zone PSA.

## 1. Introduction

Over the past 20 years, the incidence of prostate cancer (PCa) has been increasing rapidly in China [1]. Meanwhile, mortality has remained stable. A variety of reasons for the increase in incidence include longer life expectancy, changes in lifestyle, improved PCa diagnostics, etc. PCa screening using a prostate-specific antigen (PSA) results in appreciably decreased mortality and earlier diagnosis [2,3]; however, it also leads to increased unnecessary biopsies and PCa overdiagnoses [4]. To identify men who would benefit from screening, genetic assessment has become essential [5]. Recently, a tool called polygenic risk score (PRS) derived from disease-risk-associated single-nucleotide polymorphisms (SNPs) has been proven to be useful in predicting PCa inherited risk in addition to family history [6]. Our previous study also suggested that PRS could significantly predict the lifetime risk of PCa in the Chinese population [7]. It may also significantly and independently predict prostate biopsy outcomes [8,9].

On the other hand, to lower the unnecessary biopsy and increased the specificity in addition to PSA testing (high sensitivity), numerous novel approaches have been introduced in the past decade such as p2PSA and its derivative prostate health index (*phi*), urine prostate antigen 3 (PCA3), four kallikrein panel (4K score), and multiparametric magnetic resonance imaging (mpMRI) [10,11,12]. These new approaches have largely improved diagnostic accuracy and avoided unnecessary biopsies, especially in patients with gray zone PSA level (2–10 ng/mL or 4–10 ng/mL).

The incorporation between PRS and PSA for predicting PCa was reported in different studies with different populations including European [13,14], Finnish [15,16], Japanese [8], and Chinese [17]. Studies also investigated the combined effect of PRS and the novel approaches and suggested that PRS could provide significant predictive value for PCa in addition to PCA3, 4K Score, or mpMRI [6,16,17]. However, the combined effect of PRS and *phi* on PCa diagnosis has never been investigated. Therefore, we conducted the present study to evaluate the diagnostic value for PCa incorporating an East-Asian-specific PRS and *phi*.

## 2. Materials and Methods

### 2.1. Study Population

This is a prospective observational multicentric prostate biopsy cohort reported in our previous research [18]. Briefly, patients who underwent initial prostate biopsies with available genotyping information were consecutively enrolled from August 2013 to March 2019 in Shanghai, China (details in Figure 1). Notably, *phi* ((p2PSA/fPSA) × √tPSA)) was not used in clinical decision-making in this observational study. A *phi* reference with corresponding PCa risk was used in this study, similar to the one cited by most of the laboratory reports [19,20]. Blood samples were collected for genotyping and the measurement of tPSA, fPSA, and p2PSA before biopsies on the same day in a central certified lab as per the study protocol. Transrectal-ultrasound-guided systematic biopsies were performed using a 10- to 14-core scheme by the transperineal approach. All biopsy specimens were independently examined and graded by two experienced pathologists in the department of pathology at each hospital. The study was approved by the Institutional Review Board of each hospital, and written informed consent was obtained from each participant.

### 2.2. Genotyping and PRS Calculation

Genotyping was performed using the Illumina Asian Screening Array (ASA) BeadChip platform covering ≈660 k variants across the genome. With the 1000 Genomes project (East Asian population) as a reference, SNPs that were not genotyped were imputed by the Michigan Imputation Server [21]. We removed >70% of poorly imputed SNPs at the cost of <0.5% well-imputed SNPs through the Michigan Imputation Server [21].

The selection criteria of the SNPs for calculating PRS were as follows: (1) PCa-risk associated SNPs at the genome-wide significant level in East Asians from the reported GWAS studies; (2) genome-wide significant PCa-risk-associated SNPs from multiple ancestries with *p* < 0.05 and the same OR direction in East Asians. SNPs were excluded if they had (1) genotype call rate < 90%; (2) minor allele frequency (MAF) < 0.01; (3) *p* < 0.001 for the Hardy–Weinberg equilibrium (HWE) test; (4) overlapping; (5) clump-*r*^2^ > 0.1 in 250 kb clumping window. For SNPs that failed to pass quality control, we attempted to include proxy SNPs in strong linkage disequilibrium (LD, *r*^2^ > 0.8). The detailed selection procedure and criteria are shown in Figure 1.

Finally, an East-Asian-specific panel of 102 SNPs was established (Appendix A, Figure 1). A personalized PRS was calculated by summing the number of risk alleles at each SNP multiplied by the SNP’s effective size with PLINK software (version 2.0) [6]. Moreover, 10 principal components (PCs) were also calculated by it after LD pruning [22].

### 2.3. Statistical Analysis

Univariable and multivariable logistic regression was performed to test for the independent effect of factors associated with positive biopsy results and to estimate odds ratio (OR) with 95% confidence intervals (CIs). The Cochran–Armitage trend test was used to explore the trend of detection rates among groups. The discriminative performance of different biomarkers or models was evaluated by the area under the receiver operating characteristic curve (AUC) [23] and the net reclassification improvement (NRI) index [24]. A *Z*-test was used to test for the null hypothesis of NRI = 0. The final models were corrected by using repeated 10-fold cross-validation before comparison to reduce the possibility of over-fitting. All statistical analyses were performed using R version 4.1.2 [25], and a 2-tailed *p* < 0.05 was considered statistically significant.

## 3. Results

A total of 3166 patients were finally enrolled in the present study (Table 1). Among them, 1396 (44.1%) patients were diagnosed with PCa through biopsy, and 1120 (35.4%) had clinically significant PCa (Gleason score ≥ 7, csPCa). A total of 868 patients had PSA within the gray zone (2–10 ng/mL). In patients with total PSA at 2–10 ng/mL, the detection rates of PCa and csPCa were 17.6% and 15.9%, respectively. As indicated in our previous study, the detection rate of PCa was relatively low in tPSA2–20 ng/mL men [26]. We also investigated the positive biopsy rate and the performance of *phi* in men with tPSA2–20 ng/mL. The detection rate of PCa was 36.5%, and the detection rate of csPCa was 21.8%.

The median PRS in the entire cohort was 5.40 (interquartile range, IQR: 5.15–5.65). The PRS followed a normal distribution in this prostate biopsy cohort (skewness test and kurtosis test *p* > 0.05, Appendix A). Compared with men with the lowest risk in the first quintile of the PRS (0–20th PRS category), the estimated odds ratio (OR) in men with the fifth quintile of the PRS (80–100th PRS category) was 4.99 (95% CI: 3.92–6.34) (Table 2). Results were similar when adjusting for age, FH, and 10 PCs. Compared with the first quintile of PRS, the likelihood of PCa (ORs) in the second, third, fourth, and fifth quintiles of PRS were 1.86 (95%CI: 1.34–2.56), 2.07 (95%CI: 1.50–2.84), 3.26 (95%CI: 2.36–4.48), and 5.06 (95%CI:3.68–6.97), respectively (all *p* < 0.001). Similar results were observed when further adjusting for prostate volume (PV), PSA, and %fPSA (all *p* < 0.001, Table 2). The AUC of the PRS for positive biopsy results was 0.656 (95% CI: 0.637–0.675) for the entire cohort. Moreover, the predictive value of PRS for csPCa was 0.630 (AUC, 95%CI: 0.610–0.650). These implied that PRS was an independent predictive factor and suggested a supplement predictive value in addition to the FH and biomarkers.

The detection rates of PCa or csPCa based on quintiles of PRS measurements are presented in Figure 2. In the men with PSA 2–10 ng/mL, the detection rates of PCa and csPCa were 13.5% (*phi* < 27), 18.2% (*phi* 27–36), 39.7% (*phi* > 36), if further distinguished by *phi* level (Figure 2A). PRS might provide an additional predictive value for PCa in addition to *phi*. Among men with PSA 2–10 ng/mL and *phi* 27–36, the positive biopsy rate was 26.7% in men with high PRS (top 20th percentile), which was significantly higher than in those with medium PRS (20th–80th percentile, 20.3% positive rate) or low PRS (bottom 20th percentile, 8.9% positive rate, *P_trend_* = 0.03). Similarly, among men with PSA 2–10 ng/mL and *phi >* 36, the positive biopsy rates were 27.4%, 39.2%, and 58.2% in different PRS categories (*P_trend_* = 0.001). Notably, men with *phi* 27–36 but high PRS (top 20th percentile) would have a comparable risk of PCa (positive rate 26.7%) with men who had *phi* > 36 but low PRS (bottom 20th percentile). However, the trend was not significant in men with *phi* < 27. Similar results were also observed in men with PSA 2–20 ng/mL (Figure 2B). When predicting for csPCa, PRS could only distinguish = csPCa risk among men with *phi* > 36 (Figure 2C,D). In the men with PSA 2–10 ng/mL and *phi* > 36, the positive biopsy rates were 23.3%, 26.5%, and 40.0% in different PRS categories (*P_trend_* = 0.049, Figure 2C). The trend was also significant in patients with PSA 2–20 ng/mL and *phi* > 36 (*P_trend_* = 0.005, Figure 2D).

We then constructed three clinical risk models (FH-mediated model, PSA-mediated model, and *phi*-mediated model) to evaluate the predictive power of PRS on the basis of different application scenarios for PRS. The FH-mediated model included age and FH; the PSA-mediated model contained age, FH, PV, tPSA, and %fPSA; and the *phi*-mediated model did not include PSA and %fPSA but *phi*. The AUCs were 0.718 (95%CI: 0.696–0.741), 0.852 (95%CI: 0.831–0.874) and 0.904 (95%CI: 0.887–0.921) in these three models with PRS, respectively, which were significantly outperformed than models without PRS (AUC difference = 0.062, 0.012, and 0.005, respectively; all *p* < 0.05, Table 3). Moreover, adding PRSs improved reclassification with NRI ranging from 27.6% in the FH-mediated model (95%CI: 22.1–33.1%) to 13.8% in the PSA-mediated model (95%CI: 3.9–24.6%) and 8.6% in the *phi*-mediated model (95%CI: 4.2–13.0%), all *p* < 0.001 (Table 4). The NRI was even more prominent in early onset cases than in late-onset cases (44.9% vs. 26.8%, 30.8% vs. 12.0%, and 29.2% vs. 7.3% in these three models, respectively) (Table 4). This suggested that the PRS and *phi*-mediated model might be the best of the three with considerable net benefit (Appendix A). Moreover, this model fit well with the observed data (Appendix A).

## 4. Discussion

To our best knowledge, this is the first study incorporating PRS and *phi* for predicting PCa. We found that PCa-risk-associated PRS could provide a significant predictive value in addition to *phi* among men with PSA 2–10 ng/mL or 2–20 ng/mL. Firstly, PRS was independent (from FH, tPSA, *phi*, etc.) and significantly associated with PCa or csPCa according to multivariable analyses. Secondly, PRS could further distinguish the risk of PCa in men with *phi* 26–36 or *phi >* 36. To be more specific, in gray-zone PSA, men with medium *phi* (27–36) but high PRS would have a comparable risk of PCa than men with high *phi* (>36) but low PRS. Thirdly, multivariable models with PRS could provide significant net benefits (NRI), especially in early onset cases.

Both PRS and family history of PCa can measure the inherent probability of having PCa. As was shown in our study, the percentage of patients with positive family history was extremely low, largely influenced by family size, age, survival status of male relatives, and the low incidence of PCa in the Chinese population. In the multivariable model, the contribution of family history was diminished due to other strong effects of biomarkers. PRS might be a more objective and comparably suitable evaluation of PCa genetic risk than family history, although the predictive power of single PRS was limited (AUC: 0.60–0.70 in most studies) [6]. Moreover, improved assessment of PCa risk was also observed when adding PRS to a more comprehensive clinical model that includes age, prostate volume, total PSA, or *phi*, among others. In the previously reported PRS integrated model with other novel biomarkers, the AUC was 0.860 in the PCA3 + PSA + PRS model, 0.766 in the 4 K score + PRS model, and 0.784 in PSAD + PRS + DRE + age model [6,16,17]. At an intuitive level, the combined effect of PRS and *phi* was fairly outstanding. It may provide the assessment of clinical risk (biomarkers) and genetic risk as well at the same time.

One of the critical findings from the present study could be further implemented in clinical applications. As shown in the results, in the gray-zone PSA (2–10 ng/mL or 2–20 ng/mL), men with medium *phi* (27–36) but high PRS would have a comparable risk of PCa than men with high *phi* (>36) but low PRS (26.7% vs. 27.4% in men with PSA 2–10 ng/mL, 31.3% vs. 34.2% in men with PSA 2–20 ng/mL). The current clinical standard of care in China recommended that men with mild-to-moderate elevated PSA (2–10 ng/mL or 2–20 ng/mL) and high *phi* (>36) should receive an immediate biopsy. However, on the basis of the result from the current study, men with mild-to-moderate elevated PSA, as well as moderate *phi* (27–36) but high PRS (genetic risk), would have a similar risk of PCa and therefore should consider immediate biopsy as well. Notably, with additional biopsy in these men, we would be able to detect ≈3 out of 10 men with prostate biopsy, and nearly half of them would be clinically significant or high-risk PCa. Further implementation of this finding should be validated in our future study.

A strength of the study is that we have established and validated the clinical utility of a PRS that can avoid overfitting the PRS. However, several limitations need to be notified. First, the continuous follow-up for biopsy-negative patients to check for undetected cancers with repeated biopsies, as well as for biopsy-positive patients to validate the tumor stage, lymph node status, metastasis, and PCa-related death, have not yet been implemented. Second, we did not evaluate other novel biomarkers or imaging results (mpMRI) in our final analysis. Only some patients at one hospital had relevant available MRI data collected, and about 26.4% of patients took pre-biopsy mpMRI (*n* = 234). In addition, among MRI-positive patients (PIRADS ≥ 3), the percentage of high PRS (80–100%) was 25.5%, which was significantly higher than that in MRI-negative patients (15.3%, PIRADS < 3, *p* = 0.049, data not shown in the table). It is warranted to combine these clinical biomarkers and to make a comparison with more completed clinical data. Third, we were not able to provide external validation for the multivariable models but evaluated the discriminative ability with interval 10-fold validation. We expected to repeat and enrich the comprehensive clinical/genetic risk evaluation in another validation cohort in the future.

## 5. Conclusions

In summary, PRS may provide additional predictive value over *phi* for PCa. The combination of PRS and *phi* that effectively captured both clinical and genetic PCa risk is clinically practical, even in patients with gray-zone PSA.

## Figures and Tables

**Figure 1 jcm-12-01343-f001:**
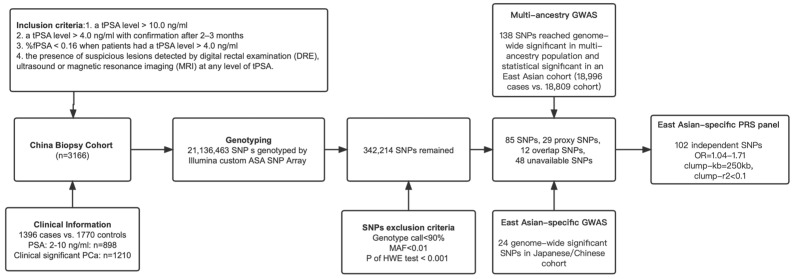
Flow diagram presenting the participant’s enrollment to the study and selection of SNPs for polygenic risk score. Abbreviation: PCa, prostate cancer; GWAS, genome-wide significant study; SNP, single-nucleotide polymorphism; ASA, Asian screening array; MAF, minor allele frequency; HWE, Hardy-Weinberg Equilibrium; PRS, polygenic risk score; tPSA, total-prostate-specific antigen; %fPSA, free PSA/tPSA; DRE, digital rectal examination; MRI, magnetic resonance imaging.

**Figure 2 jcm-12-01343-f002:**
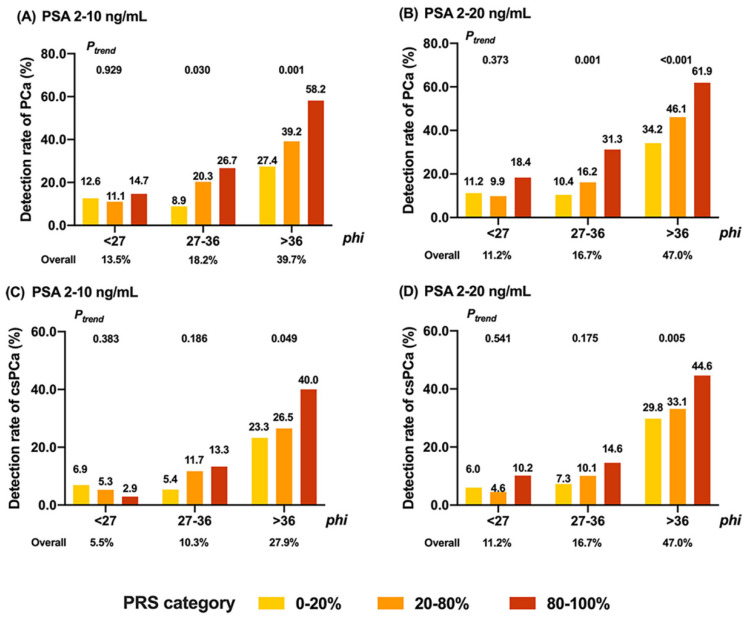
Detection rates of PCa based on quintiles of PRS in different subgroups. (**A**) The detection rates of PCa based on quintiles of PRS in PSA at 2–10 ng/mL. (**B**) The detection rates of PCa based on quintiles of PRS in PSA at 2–20 ng/mL. (**C**) The detection rates of csPCa based on quintiles of PRS in PSA at 2–10 ng/mL. (**D**) The detection rates of csPCa based on quintiles of PRS in PSA at 2–20 ng/mL. Abbreviation: PCa, prostate cancer; csPCa, clinically significant prostate cancer; PRS, polygenic risk score. Ptrend was determined by the Cochran–Armitage trend test.

**Table 1 jcm-12-01343-t001:** Baseline descriptive clinical characteristics of the China biopsy cohort.

Characteristics	All	Biopsy Positive	Biopsy Negative
Participants, no. (%)	3166	1396 (44.1)	1770 (55.9)
Age, years, median (IQR)	68 (62–74)	71 (65–77)	66 (60–72)
Family history of PCa, no. (%)	116 (3.7)	62 (4.4)	54 (3.1)
Prostate volume, mL, median (IQR)	44.13 (32.47–61.44)	38.00 (28.27–52.11)	49.45 (37.05–67.00)
Hematological tumor markers:			
Total PSA at biopsy, median (IQR)	13.02 (7.76–26.07)	22.58 (11.43–72.89)	9.73 (6.47–15.49)
Free/total PSA at biopsy, median (IQR)	0.14 (0.09–0.19)	0.12 (0.08–0.15)	0.16 (0.11–0.22)
[−2]proPSA at biopsy, median (IQR)	22.14 (12.74–53.08)	48.26 (21.33–223.52)	15.91 (10.31–24.94)
*phi* at biopsy, median (IQR)	47.39 (28.86–104.96)	102.37 (55.09–244.50)	33.08 (23.79–47.79)
Total PSA = 2–10 ng/mL, No. (%)	898 (36.1)	224 (20.3)	671 (48.3)
Total PSA = 2–20 ng/mL, No. (%)	1622 (51.2)	505 (36.2)	1117 (63.1)
Pathological results:			
Gleason score 6, No. (%)	276 (8.7)	276 (19.8)	0
Gleason score ≥ 7, No. (%)	1120 (35.4)	1120 (80.2)	0

Abbreviation: PSA, prostate-specific antigen, ng/mL; *phi*, prostate health index; IQR, interquartile range.

**Table 2 jcm-12-01343-t002:** Univariable and multivariable associations between categorized polygenic risk score and prostate cancer risk.

PRS Quintiles or Covariates ^1^	PRS Threshold	PCa Proportion, %	Unadjusted		Adjusted for Age, Family History, 10 PCs		Adjusted for Age, Family History, PSA, %fPSA, 10 PCs	
			OR (95%CI)	*p*-Value	OR (95%CI)	*p*-Value	OR (95%CI)	*p*-Value
0–20th	<5.10	25.3	1.00	Ref.	1.00	Ref.	1.00	Ref.
20–40th	5.10–5.32	37.8	1.79 (1.41–2.28)	<0.001	1.86 (1.34–2.56)	<0.001	1.40 (1.04–1.90)	0.027
40–60th	5.32–5.50	42.0	2.14 (1.69–2.72)	<0.001	2.07 (1.50–2.84)	<0.001	1.82 (1.31–2.51)	<0.001
60–80th	5.50–5.71	52.6	3.28 (2.59–4.16)	<0.001	3.26 (2.36–4.48)	<0.001	2.41 (1.75–3.32)	<0.001
80–100th	>5.71	62.8	4.99 (3.92–6.34)	<0.001	5.06 (3.68–6.97)	<0.001	3.63 (2.61–5.00)	<0.001
Age	-	-	1.07 (1.06–1.08)	<0.001	1.07 (1.06–1.08)	<0.001	1.07 (1.06–1.08)	<0.001
FH ^2^	-	-	1.60 (1.00–2.55)	0.048	1.84 (1.04–3.25)	0.036	1.80 (0.97–3.25)	0.259
PV	-	-	0.05 (0.03–0.09)	<0.001	-	-	0.01 (0.01–0.02)	<0.001
tPSA	-	-	8.45 (6.77–10.55)	<0.001	-	-	8.76 (6.55–11.82)	<0.001
%fPSA	-	-	0.12 (0.09–0.18)	<0.001	-	-	0.46 (0.30–0.72)	<0.001
*phi* ^3^	-	-	51.66 (36.80–72.53)	<0.001	-	-	38.47 (27.11–54.05)	<0.001

Abbreviation: PRS, polygenic risk score; PCa, prostate cancer; OR, odds ratio; Ref. reference; CI, confidence interval; PCs, principal components. FH, Family history of prostate cancer; PV, prostate volume; tPSA, total prostate specific antigen; %freePSA, the ratio of free PSA and total PSA; *phi*, prostate health index. ^1^ PV, PSA and *phi* value were logarithm transformed. ^2^ FH is defined as the presence or absence of at least one first- or second-degree relative with prostate cancer. ^3^
*phi*’s OR and *p* value were obtained in another model without tPSA and %fPSA.

**Table 3 jcm-12-01343-t003:** AUC differences with the addition of the PRS to different clinical risk scores.

Clinical Risk Model *	AUC	95% CI	P for AUC Comparison
FH-mediated model	0.656	0.636–0.680	Ref.
FH-mediated model + PRS	0.718	0.696–0.741	3.62 × 10^−10^
PSA-mediated model	0.840	0.818–0.863	Ref.
PSA-mediated model + PRS	0.852	0.831–0.874	3.42 × 10^−3^
*phi*-mediated model	0.899	0.881–0.916	Ref.
*phi*-mediated model + PRS	0.904	0.887–0.921	0.02

Abbreviation: FH, family history of prostate cancer; PSA, prostate-specific antigen; *phi*, prostate health index; AUC, area under ROC curves; CI, confidence interval. * Clinical risk model: FH-mediated model: Age + FH PSA-mediated model: Age + FH + log (PV) + log(tPSA) +log(%fPSA) *phi*-mediated model: Age + FH + log(PV) + log(*phi*).

**Table 4 jcm-12-01343-t004:** NRI with the addition of the PRS to different clinical risk models across age-onset groups.

Clinical Risk Model	All Individuals				Early Onset (Age ≤ 55 Year)				Late Onset (Age > 55 Year)			
	Individuals Reclassified (%)		NRI (%)		Individuals Reclassified (%)		NRI (%)		Individuals Reclassified (%)		NRI (%)	
	Up	Down	Value	95%CI	Up	Down	Value	95%CI	Up	Down	Value	95%CI
**FH-mediated model**												
Cases	38.0	24.1	13.9	8.6–19.2	50.0	23.5	26.5	−8.7~61.7	37.7	24.1	13.6	8.2–19.0
Non-cases	23.3	37.0	13.7	9.0–18.4	22.7	41.1	18.4	2.7–34.1	23.4	36.6	13.2	8.2–18.2
All	-	-	27.6	22.1–33.1	-	-	44.9	13.6–76.2	-	-	26.8	21.1–32.4
**PSA-mediated model**												
Cases	24.1	16.0	8.1	2.8–13.4	35.3	14.7	20.6	−14.5~55.7	23.9	16.5	7.4	2.0–12.8
Non-cases	17.0	22.7	5.7	1.0–10.4	12.9	22.7	10.2	−5.5~25.9	18.2	22.8	4.6	−0.4~9.6
All	-	-	13.8	3.9–24.6	-	-	30.8	4.9–55.9	-	-	12.0	7.5–16.7
** *phi* ** **-** **mediated model**												
Cases	22.6	17.4	5.2	0.1~10.5	35.3	14.7	20.6	−14.5~55.7	22.4	17.9	4.5	−0.9~9.9
Non-cases	18.1	21.5	3.4	−1.3~8.1	13.5	22.1	8.6	−7.1~24.3	19.0	21.8	2.8	−2.2~7.8
All	-	-	8.6	4.2–13.0	-	-	29.2	3.7–54.7	-	-	7.3	2.7–11.9

Abbreviation: NRI, net reclassification improvement; CI, confidence interval; PV, prostate volume; PSA, prostate-specific antigen; *phi*, prostate health index.

## Data Availability

The original contributions presented in the study are included in the article/Appendix A. Further inquiries can be directed to the corresponding authors.

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
