# Peer review of "The Combined Effect of Polygenic Risk Score and Prostate Health Index in Chinese Men Undergoing Prostate Biopsy"

_jcm, 2023, doi:10.3390/jcm12041343_

Round 1
Reviewer 1 Report
The purpose of the current manuscript was to determine the accuracy of combination between PRS and PHI index in determining the risk of developing prostate cancer. The aim of the study is clear and interesting. The topic is of major importance. However, several methodological limitations may have undermined reliability of reported results.
In particular, major drawback of the study is inherently associated to the fact that multiparametric MRI was not performed throughout prostate cancer diagnostic pathway. In addition, no data is provided in terms of clinical staging (including digital rectal examination). Finally, it is not clear which was prostate biopsy scheme adopted as well as median number of cores performed. Given this premises, all the aforementioned issues may have meaningfully influenced prostate cancer detection rate, strongly influencing the results of the study.
My advice would be to repeat the analysis by considering such elements since, as it stands, prostate cancer diagnostic pathway may appear not entirely contemporary.
Author Response
We would like to thank you for your careful and thorough consideration of our manuscript, and your enthusiasm for this work. We have substantially revised the manuscript as a result and feel our paper is greatly improved.
1) In particular, major drawback of the study is inherently associated to the fact that multiparametric MRI was not performed throughout prostate cancer diagnostic pathway.
Our response: Thanks for the concern. About 26.4% of biopsy patients in our hospital (Ruijin Hospital) took pre-biopsy mpMRI (n=234/885). In the present study, only patients undertaking systematic biopsy were enrolled and mpMRI-guided target biopsy was not included. The main purpose of the study is to investigate whether the PRS may provide extra diagnostic value in addition to the biomarker phi under this systematic biopsy cohort. Among mpMRI-positive patients (PIRADS≥3), the percentage of high PRS (80-100%) was 25.5%, which was significantly higher than that in mpMRI-negative patients (15.3%, PIRADS<3, P=0.049). This result was also added in our discussion section (lines 240-245).
On the other hand, we must emphasize that mpMRI is still not a standard-of-care procedure in many countries even in European, North America or East Asia, though it is highly recommended in the guideline. This is mainly due to the long waiting time (months to years) for mpMRI in the health systems in many countries. Therefore, from the clinical practice perspective, our results have provided additional evidence of incorporating PRS and phi for predicting PCa, which may help the vast majority of the patients at this moment.
2) In addition, no data is provided in terms of clinical staging (including digital rectal examination).
Our response: As it is a biopsy cohort, all the patients diagnosed with PCa should be graded as T1cNxM0, and only 26.4% of patients had MRI data that can provide cTNM information. It is unfortunate that we do not have enough detailed information on DRE for clinical staging but only nodule or not. Here in this study, we use the Gleason grade/score to explain the difference in disease aggressiveness among patients instead of clinical staging.
3) Finally, it is not clear which was prostate biopsy scheme adopted as well as median number of cores performed. Given this premises, all the aforementioned issues may have meaningfully influenced prostate cancer detection rate, strongly influencing the results of the study.
Our response: Thanks for your concerns. All the patients received transrectal-ultrasound-guided transperineal 10-14 core prostate biopsy. This has already been described in lines 80-84 (Method section).
Reviewer 2 Report
Dear authors
interesting work - I am not sure a PRS will provide sufficient diagnostic value to warrant the high cost associated with it, especially when compared to the OR of PHI.
additionally it is not described how many of the patients did undergo mpMRI of the prostate before biopsy - which in itself is already better than PHI.
in this regard I very much doubt there would be much benefit in a contemporary cohort in patients who underwent mpMRI and Targeted biopsy as well as systematic biopsy.
please provide the percentage of patient who underwent mpMRI of the prostate before biopsy, or if this data is not available, state so.
Non the less the paper is adequately written, with sound methods and results.
Author Response
We would like to thank you for your careful and thorough consideration of our manuscript, and your enthusiasm for this work. We have substantially revised the manuscript as a result and feel our paper is greatly improved.
It is not described how many of the patients did undergo mpMRI of the prostate before biopsy - which in itself is already better than PHI. In this regard I very much doubt there would be much benefit in a contemporary cohort in patients who underwent mpMRI and Targeted biopsy as well as systematic biopsy. Please provide the percentage of patient who underwent mpMRI of the prostate before biopsy, or if this data is not available, state so.
Our response: Thanks for the concern. About 26.4% of biopsy patients in our hospital (Ruijin Hospital) took pre-biopsy mpMRI (n=234/885). In the present study, only patients undertaking systematic biopsy were enrolled and mpMRI-guided target biopsy was not included. The main purpose of the study is to investigate whether the PRS may provide extra diagnostic value in addition to the biomarker phi under this systematic biopsy cohort. Among mpMRI-positive patients (PIRADS≥3), the percentage of high PRS (80-100%) was 25.5%, which was significantly higher than that in mpMRI-negative patients (15.3%, PIRADS<3, P=0.049). This result was also added in our discussion section.
On the other hand, we must emphasize that mpMRI is still not a standard-of-care procedure in many countries even in European, North America, or East Asia, though it is highly recommended in the guideline. This is mainly due to the long waiting time (months to years) for mpMRI in the health systems in many countries. Therefore, from the clinical practice perspective, our results have provided additional evidence of incorporating PRS and phi for predicting PCa, which may help the vast majority of the patients at this moment.
Reviewer 3 Report
Dear authors,
We thank you for the effort. With the recent knowledge on the limited effectiveness of SNP use in PCa diagnosis (see PRACTICAL Consortium papers), we must be very critical about studies such as these. You need to be careful which conclusions to make. Hence, I have some comments to hopefully improve the manuscript.
Introduction:
- “PCa screening using prostate-specific antigen (PSA) results in appreciably decreased mortality”: bold statement to make. On what do you base yourself to state this?
- “To identify men who would benefit from screening, genetic assessment has become essential”. Again, which data do you base yourself to state this? The NCCN criteria are not enough to back this up. Is there solid data showing genetic screening improves PSA based screening?
- “…increased the specificity in addition to PSA testing (high sensitivity)”. I believe sens is poor, depending on the cutoff used of course.
Methodology:
- What % of patients had an MRI pre-biopsy?
- Did all patients receive template biopsies? Targeted biopsies? In a diagnostic setting, this information is essential to exclude any bias.
Results:
- Looking at table 2. Irrespective of the PRS threshold, the % of PCa is significant. Would you omit biopsies in any of these populations based on PRS?
- Looking at figure 2, this suggests that PRS is not useful next to Phi. csPCa being the only relevant outcome, only in high Phi score you can see a difference. But you would biopsy anyways if you would have a high Phi score, so there is no added benefit of the PRS.
- The authors state that there are significant differences in the UAC with the addition with PRS, but I would need to disagree. If you compare phi versus phi + PRS. The AUC is 0.899 vs 0.904, which seems to be statistically significant, but clinically irrelevant.
- Table 4 for me is irrelevant. Too many analyses, which will ultimately lead to a significant effect. If it is not prespecified as a goal of the study, I suggest removing it.
Discussion:
The author state that they did not take MRI in consideration. What are the guidelines on the use of pre-biopsy mpMRI? In Europe and the US, this is highly recommended. The question than remains what the added benefit of the PRS (and Phi) is of course
Author Response
We would like to thank you for your careful and thorough consideration of our manuscript, and your enthusiasm for this work. We have substantially revised the manuscript as a result and feel our paper is greatly improved.
Introduction:
1) “PCa screening using prostate-specific antigen (PSA) results in appreciably decreased mortality”: bold statement to make. On what do you base yourself to state this?
Our response: Evidence from the random Göteborg-1 trials has demonstrated a 44% reduction in prostate cancer-specific mortality at 14 yr of follow-up in favor of PSA screening [1]. With the extended follow-up to 22yr, the mortality reduction is sustained, with a relative risk reduction of 29% [2]. Based on this trial, we state so and cited these two references in the main text.
Reference:
[1] Hugosson J, et al. Mortality results from the Göteborg randomized population-based prostate-cancer screening trial. Lancet Oncol. 2010 Aug;11(8):725-32. doi: 10.1016/S1470-2045(10)70146-7.
[2] Frånlund M, et al. Results from 22 years of Followup in the Göteborg Randomized Population-Based Prostate Cancer Screening Trial. J Urol. 2022 Aug;208(2):292-300. doi: 10.1097/JU.0000000000002696.
2) “To identify men who would benefit from screening, genetic assessment has become essential”. Again, which data do you base yourself to state this? The NCCN criteria are not enough to back this up. Is there solid data showing genetic screening improves PSA based screening?
Our response: First of all, the genetic assessment did not only contain PRS or genetic risk score, but also the genetic background, family history, and germline pathogenic mutations (for instance, BRCA2, HOXB13). Besides genotype-adjusted PSA used in personalized PSA-based screening, genetic risk stratification by PRS or others could also improve it [1]. Based on the result of the Stockholm-1 cohort study. Aly M also proved that the genetic model corporating with PRS would have reduced the number of biopsies carried out by 22.7% compared with the clinical model (which recommended biopsy in all subjects) [2]. Above all, the genetic assessment would improve the accuracy of serum PSA screening for prostate cancer from different aspects. More importantly, such a strategy has also been recommended by other clinical guidelines, including EAU-EANM-ESTRO-ESUR-ISUP-SIOG guidelines on prostate cancer.
[1] Scardino PT. Prostate cancer: improving PSA testing by adjusting for genetic background. Nat Rev Urol. 2013 Apr;10(4):190-2. doi: 10.1038/nrurol.
[2] Aly M, Wiklund F, Xu J, Isaacs WB, Eklund M, D'Amato M, Adolfsson J, Grönberg H. Polygenic risk score improves prostate cancer risk prediction: results from the Stockholm-1 cohort study. Eur Urol. 2011 Jul;60(1):21-8. doi: 10.1016/j.eururo.2011.01.017.
3) “…increased the specificity in addition to PSA testing (high sensitivity)”. I believe sens is poor, depending on the cutoff used of course.
Our response: The level of sensitivity depends on the cutoff. As a matter of fact, under the most commonly used and the FDA-recommended cutoff value of 4ng/mL, the sensitivity of PSA was 86% and specificity was 33%. Results reported by other groups in our institute also suggested that the sensitivity ranged from 99.7% to 92.4% and specificity ranged from 4.4% to 37.3% under the cutoff value of 4-10ng/mL[1].
Reference:
[1] Na R, Jiang H, Kim ST, Wu Y, Tong S, Zhang L, Xu J, Sun Y, Ding Q. Outcomes and trends of prostate biopsy for prostate cancer in Chinese men from 2003 to 2011. PLoS One. 2012;7(11):e49914.
Methodology:
4) What % of patients had an MRI pre-biopsy?
Our response: About 26.4% of biopsy patients in our hospital (Ruijin Hospital) took pre-biopsy mpMRI (n=234/885). In the present study, only patients undertaking systematic biopsy were enrolled and mpMRI-guided target biopsy was not included.
5) Did all patients receive template biopsies? Targeted biopsies? In a diagnostic setting, this information is essential to exclude any bias.
Our response: All patients received ultrasound-guided transperineal 10-14core prostate biopsy. Please refer to lines 80-84.
6) Looking at table 2. Irrespective of the PRS threshold, the % of PCa is significant. Would you omit biopsies in any of these populations based on PRS?
Our response: In the Chinese population, as reported by another study, the positive biopsy rates ranged from 74% in 2003 to 33% in 2011 (around 40%) [1-3]. Among patients with PSA 2-10ng/mL, the positive biopsy rates ranged from 17.6% to 20.6% [2-3]. A similar positive rate was observed in our cohort. In fact, the key message from this study is that among patients with mildly elevated phi (27-36, not recommend immediate biopsy), those who have high genetic risk (based on PRS) should also recommend an immediate biopsy (Figure 2, result section line 150-163). We did not recommend any delayed biopsy based on PRS but recommended possible valid biopsies based on the results from the current study.
Reference:
[1] Na R, Jiang H, Kim ST, Wu Y, Tong S, Zhang L, Xu J, Sun Y, Ding Q. Outcomes and trends of prostate biopsy for prostate cancer in Chinese men from 2003 to 2011. PLoS One. 2012;7(11):e49914.
[2] Na R, Ye D, Liu F, Chen H, Qi J, Wu Y, Zhang G, Wang M, Wang W, Sun J, Yu G, Zhu Y, Ren S, Zheng SL, Jiang H, Sun Y, Ding Q, Xu J. Performance of serum prostate-specific antigen isoform [-2]proPSA (p2PSA) and the prostate health index (PHI) in a Chinese hospital-based biopsy population. Prostate. 2014 Nov;74(15):1569-75.
[3] Na R, Ye D, Qi J, Liu F, Helfand BT, Brendler CB, Conran CA, Packiam V, Gong J, Wu Y, Zheng SL, Mo Z, Ding Q, Sun Y, Xu J. Prostate health index significantly reduced unnecessary prostate biopsies in patients with PSA 2-10 ng/mL and PSA >10 ng/mL: Results from a Multicenter Study in China. Prostate. 2017 Aug;77(11):1221-1229.
7) Looking at figure 2, this suggests that PRS is not useful next to Phi. csPCa being the only relevant outcome, only in high Phi score you can see a difference. But you would biopsy anyways if you would have a high Phi score, so there is no added benefit of the PRS.
Our response: Please refer to the part of discussion lines 232-241. As shown in figure 2, men with mild-to-moderate elevated PSA, and moderate phi (27-36) but high PRS (genetic risk) would have a similar risk of PCa. This would help to decide this subset of patients whether to be biopsied, which was the critical point of finding.
8) The authors state that there are significant differences in the AUC with the addition with PRS, but I would need to disagree. If you compare phi versus phi + PRS. The AUC is 0.899 vs 0.904, which seems to be statistically significant, but clinically irrelevant. Table 4 for me is irrelevant. Too many analyses, which will ultimately lead to a significant effect. If it is not prespecified as a goal of the study, I suggest removing it.
Our response: We agreed that only a very mildly improved AUC was observed. However, AUC is one of the many approaches to evaluate the diagnostic value. Its limitations have been mentioned in the reported studies (cite: https://en.wikipedia.org/wiki/Net_reclassification_improvement ref #2 #3). In addition to the AUC, net reclassification improvement is another valuable tool to evaluate the magnitude of improvement. As listed in Table 4, The NRI was even more prominent in early-onset cases than in late-onset cases (44.9% vs. 26.8%, 30.8% vs. 12.0%, and 29.2% vs. 7.3% in these three models, respectively), which emphasized the role of PRS on indicating disease status under prostate biopsy among young patients. So we still want to keep table 4 to illustrate this point.
9) The author state that they did not take MRI in consideration. What are the guidelines on the use of pre-biopsy mpMRI? In Europe and the US, this is highly recommended. The question than remains what the added benefit of the PRS (and Phi) is of course
Our response: Thank you for your concerns. we must emphasize that mpMRI is still not a standard-of-care procedure in many countries even in European, North America, or East Asia, though it is highly recommended in the guideline. This is mainly due to the long waiting time (months to years) for mpMRI in the health systems in many countries. Therefore, from the clinical practice perspective, our results have provided additional evidence of incorporating PRS and phi for predicting PCa, which may help the vast majority of the patients at this moment. The main purpose of the study is to investigate whether the PRS may provide extra diagnostic value in addition to the biomarker phi under this systematic biopsy cohort. Among mpMRI-positive patients (PIRADS≥3), the percentage of high PRS (80-100%) was 25.5%, which was significantly higher than that in mpMRI-negative patients (15.3%, PIRADS<3, P=0.049). This result was also added in our results section.